# Learning feed-forward one-shot learners

**Luca Bertinetto**[*]
University of Oxford
luca@robots.ox.ac.uk

**João F. Henriques**[*]
University of Oxford
joao@robots.ox.ac.uk

**Jack Valmadre**[*]
University of Oxford
jvlmdr@robots.ox.ac.uk

**Philip H. S. Torr**
University of Oxford
philip.torr@eng.ox.ac.uk

**Andrea Vedaldi**
University of Oxford
vedaldi@robots.ox.ac.uk

## Abstract

One-shot learning is usually tackled by using generative models or discriminative embeddings. Discriminative methods based on deep learning, which are very effective in other learning scenarios, are ill-suited for one-shot learning as they need large amounts of training data. In this paper, we propose a method to learn the parameters of a deep model in one shot. We construct the learner as a second deep network, called a *learnet*, which predicts the parameters of a pupil network from a single exemplar. In this manner we obtain an efficient feed-forward one-shot learner, trained end-to-end by minimizing a one-shot classification objective in a learning to learn formulation. In order to make the construction feasible, we propose a number of factorizations of the parameters of the pupil network. We demonstrate encouraging results by learning characters from single exemplars in Omniglot, and by tracking visual objects from a single initial exemplar in the Visual Object Tracking benchmark.

## 1 Introduction

Deep learning methods have taken by storm areas such as computer vision, natural language processing and speech recognition. One of their key strengths is the ability to leverage large quantities of labelled data and extract meaningful and powerful representations from it. However, this capability is also one of their most significant limitations since using large datasets to train deep neural network is not just an option, but a necessity. It is well known, in fact, that these models are prone to overfitting.

Thus, deep networks seem less useful when the goal is to learn a new concept on the fly, from a few or even a single example as in one shot learning. These problems are usually tackled by using generative models [18, 13] or, in a discriminative setting, using ad-hoc solutions such as exemplar support vector machines (SVMs) [14]. Perhaps the most common discriminative approach to one-shot learning is to learn off-line a deep *embedding function* and then to define on-line simple classification rules such as nearest neighbors in the embedding space [5, 16]. However, computing an embedding is a far cry from learning a model of the new object.

In this paper, we take a very different approach and ask whether we can induce, from a single supervised example, a *full, deep discriminative model* to recognize other instances of the same object class. Furthermore, we do not want our solution to require a lengthy optimization process, but to be computable on-the-fly, efficiently and in one go. We formulate this problem as the one of learning a deep neural network, called a *learnet*, that, given a single exemplar of a new object class, predicts the parameters of a second network that can recognize other objects of the same type.

---

[*]The first three authors contributed equally, and are listed in alphabetical order.

Our model has several elements of interest. Firstly, if we consider learning to be any process that maps a set of images to the parameters of a model, then it can be seen as a "learning to learn" approach. Clearly, learning from a single exemplar is only possible given sufficient prior knowledge on the learning domain. This prior knowledge is incorporated in the learnet in an off-line phase by solving millions of small one-shot learning tasks and back-propagating errors end-to-end. Secondly, our learnet provides a feed-forward learning algorithm that extracts from the available exemplar the final model parameters in one go. This is different from iterative approaches such as exemplar SVMs or complex inference processes in generative modeling. It also demonstrates that deep neural networks can learn at the "meta-level" of predicting filter parameters for a second network, which we consider to be an interesting result in its own right. Thirdly, our method provides a competitive, efficient, and practical way of performing one-shot learning using discriminative methods.

## 1.1 Related work

Our work is related to several others in the literature. However, we believe to be the first to look at methods that can learn the parameters of complex discriminative models in one shot.

One-shot learning has been widely studied in the context of generative modeling, which unlike our work is often *not* focused on solving discriminative tasks. One very recent example is by Rezende et al. [18], which uses a recurrent spatial attention model to generate images, and learns by optimizing a measure of reconstruction error using variational inference [9]. They demonstrate results by sampling images of novel classes from this generative model, not by solving discriminative tasks. Another notable work is by Lake et al. [13], which instead uses a probabilistic program as a generative model. This model constructs written characters as compositions of pen strokes, so although more general programs can be envisioned, they demonstrate it only on Optical Character Recognition (OCR) applications.

A different approach to one-shot learning is to learn an embedding space, which is typically done with a siamese network [2]. Given an exemplar of a novel category, classification is performed in the embedding space by a simple rule such as nearest-neighbor. Training is usually performed by classifying pairs according to distance [5], or by enforcing a distance ranking with a triplet loss [16].

Our work departs from the paradigms of generative modeling and similarity learning, instead predicting the parameters of a neural network from a single exemplar image. It can be seen as a network that effectively "learns to learn", generalizing across tasks defined by different exemplars.

The idea of parameter prediction was, to our knowledge, first explored by Schmidhuber [20] in a recurrent architecture with one network that modifies the weights of another. Parameter prediction has also been used for zero-shot learning (as opposed to one-shot learning), which is the related problem of learning a new object class without a single example image, based solely on a description such as binary attributes or text. Whereas it is usually framed as a modality transfer problem and solved through transfer learning [21], Noh et al. [15] recently employed parameter prediction to induce the weights of an image classifier from text for the problem of visual question answering.

Denil et al. [4] investigated the redundancy of neural network parameters, showing that it is possible to linearly predict as many as 95% of the parameters in a layer given the remaining 5%. This is a vastly different proposition from ours, which is to predict *all* of the parameters of a layer given an external exemplar image, and to do so non-linearly.

## 2 One-shot learning as dynamic parameter prediction

Since we consider one-shot learning as a discriminative task, our starting point is standard discriminative learning. It generally consists of finding the parameters $W$ that minimize the average loss $\mathcal{L}$ of a predictor function $\varphi(x; W)$, computed over a dataset of $n$ samples $x_i$ and corresponding labels $\ell_i$:

$$\min_{W} \frac{1}{n} \sum_{i=1}^{n} \mathcal{L}(\varphi(x_i; W), \ell_i). \tag{1}$$

Unless the model space is very small, generalization also requires constraining the choice of model, usually via regularization. However, in the extreme case in which the goal is to learn $W$ from a single exemplar $z$ of the class of interest, called *one-shot learning*, even regularization may be insufficient and additional prior information must be injected into the learning process. The main challenge in

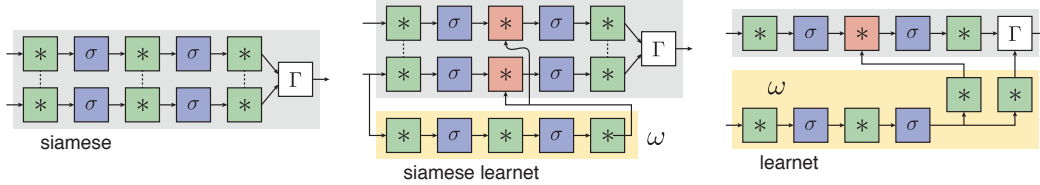

Figure 1: Our proposed architectures predict the parameters of a network from a single example, replacing static convolutions (green) with dynamic convolutions (red). The siamese learnet predicts the parameters of an embedding function that is applied to both inputs, whereas the single-stream learnet predicts the parameters of a function that is applied to the other input. Linear layers are denoted by $*$ and nonlinear layers by $\sigma$. Dashed connections represent parameter sharing.

discriminative one-shot learning is to find a mechanism to incorporate domain-specific information in the learner, i.e. *learning to learn*. Another challenge, which is of practical importance in applications of one-shot learning, is to avoid a lengthy optimization process such as eq. (1).

We propose to address both challenges by learning the parameters $W$ of the predictor from a single exemplar $z$ using a meta-prediction process, i.e. a non-iterative feed-forward function $\omega$ that maps $(z; W')$ to $W$. Since in practice this function will be implemented using a deep neural network, we call it a *learnet*. The learnet depends on the exemplar $z$, which is a single representative of the class of interest, and contains parameters $W'$ of its own. Learning to learn can now be posed as the problem of optimizing the learnet meta-parameters $W'$ using an objective function defined below. Furthermore, the feed-forward learnet evaluation is much faster than solving the optimization problem (1).

In order to train the learnet, we require the latter to produce good predictors given any possible exemplar $z$, which is empirically evaluated as an average over $n$ training samples $z_i$:

$$\min_{W'} \frac{1}{n} \sum_{i=1}^{n} \mathcal{L}(\varphi(x_i; \omega(z_i; W')), \ell_i). \tag{2}$$

In this expression, the performance of the predictor extracted by the learnet from the exemplar $z_i$ is assessed on a single "validation" pair $(x_i, \ell_i)$, comprising another exemplar and its label $\ell_i$. Hence, the training data consists of triplets $(x_i, z_i, \ell_i)$. Notice that the meaning of the label $\ell_i$ is subtly different from eq. (1) since the class of interest changes depending on the exemplar $z_i$: $\ell_i$ is positive when $x_i$ and $z_i$ belong to the same class and negative otherwise. Triplets are sampled uniformly with respect to these two cases. Importantly, the parameters of the original predictor $\varphi$ of eq. (1) now change dynamically with each exemplar $z_i$.

Note that the training data is reminiscent of that of siamese networks [2], which also learn from labeled sample pairs. However, siamese networks apply the same model $\varphi(x; W)$ with shared weights $W$ to both $x_i$ and $z_i$, and compute their inner-product to produce a similarity score:

$$\min_{W} \frac{1}{n} \sum_{i=1}^{n} \mathcal{L}(\langle \varphi(x_i; W), \varphi(z_i; W) \rangle, \ell_i). \tag{3}$$

There are two key differences with our model. First, we treat $x_i$ and $z_i$ asymmetrically, which results in a different objective function. Second, and most importantly, the output of $\omega(z; W')$ is used to parametrize linear layers that determine the intermediate representations in the network $\varphi$. This is significantly different to computing a single inner product in the last layer (eq. (3)).

Eq. (2) specifies the optimization objective of one-shot learning as dynamic parameter prediction. By application of the chain rule, backpropagating derivatives through the computational blocks of $\varphi(x; W)$ and $\omega(z; W')$ is no more difficult than through any other standard deep network. Nevertheless, when we dive into concrete implementations of such models we face a peculiar challenge, discussed next.

## 2.1 The challenge of naive parameter prediction

In order to analyse the practical difficulties of implementing a learnet, we will begin with one-shot prediction of a fully-connected layer, as it is simpler to analyse. This is given by

$$y = Wx + b, \tag{4}$$

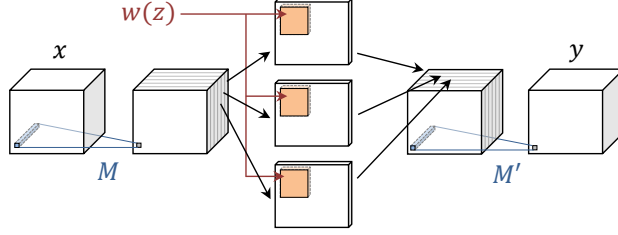

Figure 2: Factorized convolutional layer (eq. (8)). The channels of the input $x$ are projected to the factorized space by $M$ (a $1 \times 1$ convolution), the resulting channels are convolved independently with a corresponding filter prediction from $w(z)$, and finally projected back using $M'$.

given an input $x \in \mathbb{R}^d$, output $y \in \mathbb{R}^k$, weights $W \in \mathbb{R}^{k \times d}$ and biases $b \in \mathbb{R}^k$.

We now replace the weights and biases with their functional counterparts, $w(z)$ and $b(z)$, representing two outputs of the learnet $\omega(z; W')$ given the exemplar $z \in \mathbb{R}^m$ as input (to avoid clutter, we omit the implicit dependence on $W'$):

$$y = w(z)x + b(z). \tag{5}$$

While eq. (5) seems to be a drop-in replacement for linear layers, careful analysis reveals that it scales extremely poorly. The main cause is the unusually large output space of the learnet $w : \mathbb{R}^m \to \mathbb{R}^{k \times d}$. For a comparable number of input and output units in a linear layer ($d \simeq k$), the output space of the learnet grows *quadratically* with the number of units.

While this may seem to be a concern only for large networks, it is actually extremely difficult also for networks with few units. Consider a simple linear learnet $w(z) = W'z$. Even for a very small fully-connected layer of only 100 units ($d = k = 100$), and an exemplar $z$ with 100 features ($m = 100$), the learnet already contains 1M parameters that must be learned. Overfitting and space and time costs make learning such a regressor infeasible. Furthermore, reducing the number of features in the exemplar can only achieve a small constant-size reduction on the total number of parameters. The bottleneck is the quadratic size of the *output space* $dk$, not the size of the input space $m$.

## 2.2 Factorized linear layers

A simple way to reduce the size of the output space is to consider a factorized set of weights, by replacing eq. (5) with:

$$y = M' \operatorname{diag}(w(z)) Mx + b(z). \tag{6}$$

The product $M'\operatorname{diag}(w(z)) M$ can be seen as a factorized representation of the weights, analogous to the Singular Value Decomposition. The matrix $M \in \mathbb{R}^{d \times d}$ projects $x$ into a space where the elements of $w(z)$ represent disentangled factors of variation. The second projection $M' \in \mathbb{R}^{k \times d}$ maps the result back from this space.

Both $M$ and $M'$ contain additional parameters to be learned, but they are modest in size compared to the case discussed in sect. 2.1. Importantly, the one-shot branch $w(z)$ now only has to predict a set of diagonal elements (see eq. (6)), so its output space grows linearly with the number of units in the layer (i.e. $w(z): \mathbb{R}^m \to \mathbb{R}^d$).

## 2.3 Factorized convolutional layers

The factorization of eq. (6) can be generalized to convolutional layers as follows. Given an input tensor $x \in \mathbb{R}^{r \times c \times d}$, weights $W \in \mathbb{R}^{f \times f \times d \times k}$ (where $f$ is the filter support size), and biases $b \in \mathbb{R}^k$, the output $y \in \mathbb{R}^{r' \times c' \times k}$ of the convolutional layer is given by

$$y = W * x + b, \tag{7}$$

where $*$ denotes convolution, and the biases $b$ are applied to each of the $k$ channels.

Projections analogous to $M$ and $M'$ in eq. (6) can be incorporated in the filter bank in different ways and it is not obvious which one to pick. Here we take the view that $M$ and $M'$ should disentangle the feature channels (i.e. third dimension of $x$) so that the predicted filters $w(z)$ can operate on each channel independently. As such, we consider the following factorization:

$$y = M' * w(z) *_d M * x + b(z), \tag{8}$$

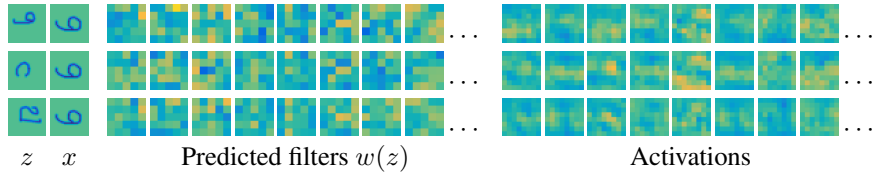

Figure 3: The predicted filters and the output of a dynamic convolutional layer in a single-stream learnet trained for the OCR task. Different exemplars $z$ define different filters $w(z)$. Applying the filters of each exemplar to the same input $x$ yields different responses. Best viewed in colour.

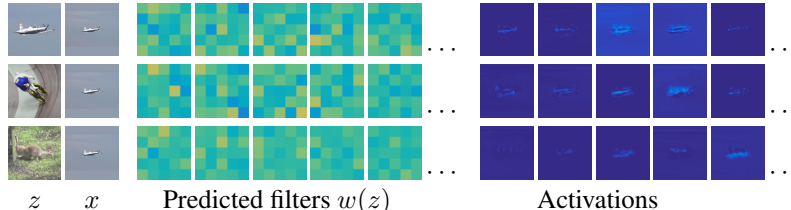

Figure 4: The predicted filters and the output of a dynamic convolutional layer in a siamese learnet trained for the object tracking task. Best viewed in colour.

where $M \in \mathbb{R}^{1 \times 1 \times d \times d}$, $M' \in \mathbb{R}^{1 \times 1 \times d \times k}$, and $w(z) \in \mathbb{R}^{f \times f \times d}$. Convolution with subscript $d$ denotes independent filtering of $d$ channels, i.e. each channel of $x *_d y$ is simply the convolution of the corresponding channel in $x$ and $y$. In practice, this can be achieved with filter tensors that are diagonal in the third and fourth dimensions, or using $d$ filter groups [12], each group containing a single filter. An illustration is given in fig. 2. The predicted filters $w(z)$ can be interpreted as a filter basis, as described in the supplementary material (sec. A).

Notice that, under this factorization, the number of elements to be predicted by the one-shot branch $w(z)$ is only $f^2 d$ (the filter size $f$ is typically very small, e.g. 3 or 5 [5, 23]). Without the factorization, it would be $f^2 dk$ (the number of elements of $W$ in eq. (7)). Similarly to the case of fully-connected layers (sect. 2.2), when $d \simeq k$ this keeps the number of predicted elements from growing quadratically with the number of channels, allowing them to grow only linearly.

Examples of filters that are predicted by learnets are shown in figs. 3 and 4. The resulting activations confirm that the networks induced by different exemplars do indeed possess different internal representations of the same input.

## 3 Experiments

We evaluate learnets against baseline one-shot architectures (sect. 3.1) on two one-shot learning problems in Optical Character Recognition (OCR; sect. 3.2) and visual object tracking (sect. 3.3). All experiments were performed using MatConvNet [22].

### 3.1 Architectures

As noted in sect. 2, the closest competitors to our method in discriminative one-shot learning are embedding learning using siamese architectures. Therefore, we structure the experiments to compare against this baseline. In particular, we choose to implement learnets using similar network topologies for a fairer comparison.

The baseline siamese architecture comprises two parallel streams $\varphi(x; W)$ and $\varphi(z; W)$ composed of a number of layers, such as convolution, max-pooling, and ReLU, sharing parameters $W$ (fig. 1.a). The outputs of the two streams are compared by a layer $\Gamma(\varphi(x; W), \varphi(z; W))$ computing a measure of similarity or dissimilarity. We consider in particular: the dot product $\langle a, b \rangle$ between vectors $a$ and $b$, the Euclidean distance $\|a - b\|$, and the weighted $l^1$-norm $\|w \odot a - w \odot b\|_1$ where $w$ is a vector of learnable weights and $\odot$ the Hadamard product).

The first modification to the siamese baseline is to use a learnet to predict some of the intermediate shared stream parameters (fig. 1.b). In this case $W = \omega(z; W')$ and the siamese architecture writes $\Gamma(\varphi(x; \omega(z; W')), \varphi(z; \omega(z; W')))$. Note that the siamese parameters are still the same in the two

Table 1: Error rate for character recognition in foreign alphabets (chance is 95%).

| | Inner-product (%) | Euclidean dist. (%) | Weighted $\ell^1$ dist. (%) |
|---|---|---|---|
| Siamese (shared) | 48.5 | 37.3 | 41.8 |
| Siamese (unshared) | 47.0 | 41.0 | 34.6 |
| Siamese (unshared, fact.) | 48.4 | – | 33.6 |
| Siamese learnet (shared) | 51.0 | 39.8 | 31.4 |
| Learnet | 43.7 | 36.7 | **28.6** |
| Modified Hausdorff distance | | 43.2 | |

streams, whereas the learnet is an entirely new subnetwork whose purpose is to map the exemplar image to the shared weights. We call this model the *siamese learnet*.

The second modification is a *single-stream learnet* configuration, using only one stream $\varphi$ of the siamese architecture and predicting its parameter using the learnet $\omega$. In this case, the comparison block $\Gamma$ is reinterpreted as the last layer of the stream $\varphi$ (fig. 1.c). Note that: i) the single predicted stream and learnet are asymmetric and with different parameters and ii) the learnet predicts both the final comparison layer parameters $\Gamma$ as well as intermediate filter parameters.

The single-stream learnet architecture can be understood to predict a discriminant function from one example, and the siamese learnet architecture to predict an embedding function for the comparison of two images. These two variants demonstrate the versatility of the dynamic convolutional layer from eq. (6).

Finally, in order to ensure that any difference in performance is not simply due to the asymmetry of the learnet architecture or to the induced filter factorizations (sect. 2.2 and sect. 2.3), we also compare *unshared* siamese nets, which use distinct parameters for each stream, and *factorized* siamese nets, where convolutions are replaced by factorized convolutions as in learnet.

## 3.2 Character recognition in foreign alphabets

This section describes our experiments in one-shot learning on OCR. For this, we use the Omniglot dataset [13], which contains images of handwritten characters from 50 different alphabets. These alphabets are divided into 30 *background* and 20 *evaluation* alphabets. The associated one-shot learning problem is to develop a method for determining whether, given any single exemplar of a character in an evaluation alphabet, any other image in that alphabet represents the same character or not. Importantly, all methods are trained using only background alphabets and tested on the evaluation alphabets.

**Dataset and evaluation protocol.** Character images are resized to $28 \times 28$ pixels in order to be able to explore efficiently several variants of the proposed architectures. There are exactly 20 sample images for each character, and an average of 32 characters per alphabet. The dataset contains a total of 19,280 images in the background alphabets and 13,180 in the evaluation alphabets.

Algorithms are evaluated on a series of recognition problems. Each recognition problem involves identifying the image in a set of 20 that shows the same character as an exemplar image (there is always exactly one match). All of the characters in a single problem belong to the same alphabet. At test time, given a collection of characters $(x_1, \ldots, x_m)$, the function is evaluated on each pair $(z, x_i)$ and the candidate with the highest score is declared the match. In the case of the learnet architectures, this can be interpreted as obtaining the parameters $W = \omega(z; W')$ and then evaluating a static network $\varphi(x_i; W)$ for each $x_i$.

**Architecture.** The baseline stream $\varphi$ for the siamese, siamese learnet, and single-stream learnet architecture consists of 3 convolutional layers, with $2 \times 2$ max-pooling layers of stride 2 between them. The filter sizes are $5 \times 5 \times 1 \times 16$, $5 \times 5 \times 16 \times 64$ and $4 \times 4 \times 64 \times 512$. For both the siamese learnet and the single-stream learnet, $\omega$ consists of the same layers as $\varphi$, except the number of outputs is 1600 – one for each element of the 64 predicted filters (of size $5 \times 5$). To keep the experiments simple, we only predict the parameters of one convolutional layer. We conducted cross-validation to choose the predicted layer and found that the second convolutional layer yields the best results for both of the proposed variants.

Siamese nets have previously been applied to this problem by Koch et al. [10] using much deeper networks applied to images of size $105 \times 105$. However, we have restricted this investigation to relatively shallow networks to enable a thorough exploration of the parameter space. A more powerful

algorithm for one-shot learning, Hierarchical Bayesian Program Learning [13], is able to achieve human-level performance. However, this approach involves computationally expensive inference at test time, and leverages extra information at training time that describes the strokes drawn by the human author.

**Learning.** Learning involves minimizing the objective function specific to each method (e.g. eq. (2) for learnet and eq. (3) for siamese architectures) and uses stochastic gradient descent (SGD) in all cases. As noted in sect. 2, the objective is obtained by sampling triplets $(z_i, x_i, \ell_i)$ where exemplars $z_i$ and $x_i$ are congruous ($\ell_i = +1$) or incongruous ($\ell_i = -1$) with 50% probability. We consider 100,000 random pairs for training per epoch, and train for 60 epochs. We conducted a random search to find the best hyper-parameters for each algorithm (initial learning rate and geometric decay, standard deviation of Gaussian parameter initialization, and weight decay).

**Results and discussion.** Tab. 1 shows the classification error obtained using variants of each architecture. A dash indicates a failure to converge given a large range of hyper-parameters. The two learnet architectures combined with the weighted $\ell^1$ distance are able to achieve significantly better results than other methods. The best architecture reduced the error from 37.3% for a siamese network with shared parameters to 28.6% for a single-stream learnet.

While the Euclidean distance gave the best results for siamese networks with shared parameters, better results were achieved by learnets (and siamese networks with unshared parameters) using a weighted $\ell^1$ distance. In fact, none of the alternative architectures are able to achieve lower error under the Euclidean distance than the shared siamese net. The dot product was, in general, less effective than the other two metrics.

The introduction of the factorization in the convolutional layer might be expected to improve the quality of the estimated model by reducing the number of parameters, or to worsen it by diminishing the capacity of the hypothesis space. For this relatively simple task of character recognition, the factorization did not seem to have a large effect.

## 3.3 Object tracking

The task of single-target object tracking requires to locate an object of interest in a sequence of video frames. A video frame can be seen as a collection $\mathcal{F} = \{w_1, \ldots, w_K\}$ of image windows; then, in a one-shot setting, given an exemplar $z \in \mathcal{F}_1$ of the object in the first frame $\mathcal{F}_1$, the goal is to identify the same window in the other frames $\mathcal{F}_2, \ldots, \mathcal{F}_M$.

**Datasets.** The method is trained using the ImageNet Large Scale Visual Recognition Challenge 2015 [19], with 3,862 videos totalling more than one million annotated frames. Instances of objects of thirty different classes (mostly vehicles and animals) are annotated throughout each video with bounding boxes. For tracking, instance labels are retained but object class labels are ignored. We use 90% of the videos for training, while the other 10% are held-out to monitor validation error during network training. Testing uses the VOT 2015 benchmark [11].

**Architecture.** We experiment with siamese and siamese learnet architectures (fig. 1) where the learnet $\omega$ predicts the parameters of the second (dynamic) convolutional layer of the siamese streams. Each siamese stream has five convolutional layers and we test three variants of those: variant (A) has the same configuration as AlexNet [12] but with stride 2 in the first layer, and variants (B) and (C) reduce to 50% the number of filters in the first two convolutional layers and, respectively, to 25% and 12.5% the number of filters in the last layer.

**Training.** In order to train the architecture efficiently from many windows, the data is prepared as follows. Given an object bounding box sampled at random, a crop $z$ double the size of that is extracted from the corresponding frame, padding with the average image color when needed. The border is included in order to incorporate some visual context around the exemplar object. Next, $\ell \in \{+1, -1\}$ is sampled at random with 75% probability of being positive. If $\ell = -1$, an image $x$ is extracted by choosing at random a frame that does *not* contain the object. Otherwise, a second frame containing the same object and within 50 temporal samples of the first is selected at random. From that, a patch $x$ centered around the object and four times bigger is extracted. In this way, $x$ contains both subwindows that do and do not match $z$. Images $z$ and $x$ are resized to $127 \times 127$ and $255 \times 255$ pixels, respectively, and the triplet $(z, x, \ell)$ is formed. All $127 \times 127$ subwindows in $x$ are considered to *not* match $z$ except for the central $2 \times 2$ ones when $\ell = +1$.

Table 2: Tracking accuracy and number of tracking failures in the VOT 2015 Benchmark, as reported by the toolkit [11]. Architectures are grouped by size of the main network (see text). For each group, the best entry for each column is in bold. We also report the scores of 5 recent trackers.

| Method | Accuracy | Failures | Method | Accuracy | Failures |
|---|---|---|---|---|---|
| Siamese ($\varphi$=B) | 0.465 | 105 | Siamese ($\varphi$=C) | 0.466 | 120 |
| Siamese ($\varphi$=B; unshared) | 0.447 | 131 | Siamese ($\varphi$=C; factorized) | 0.435 | 132 |
| Siamese ($\varphi$=B; factorized) | 0.444 | 138 | Siamese learnet ($\varphi$=C; $\omega$=A) | 0.483 | **105** |
| Siamese learnet ($\varphi$=B; $\omega$=A) | **0.500** | **87** | Siamese learnet ($\varphi$=C; $\omega$=C) | **0.491** | 106 |
| Siamese learnet ($\varphi$=B; $\omega$=B) | 0.497 | 93 | DSST [3] | 0.483 | 163 |
| DAT [17] | 0.442 | 113 | MEEM [24] | 0.458 | 107 |
| SO-DLT [23] | 0.540 | 108 | MUSTer [7] | 0.471 | 132 |

All networks are trained from scratch using SGD for 50 epoch of 50,000 sample triplets $(z_i, x_i, \ell_i)$. The multiple windows contained in $x$ are compared to $z$ efficiently by making the comparison layer $\Gamma$ convolutional (fig. 1), accumulating a logistic loss across spatial locations. The same hyperparameters (learning rate of $10^{-2}$ geometrically decaying to $10^{-5}$, weight decay of 0.005, and small mini-batches of size 8) are used for all experiments, which we found to work well for both the baseline and proposed architectures. The weights are initialized using the improved Xavier [6] method, and we use batch normalization [8] after all linear layers.

**Testing.** Adopting the initial crop as exemplar, the object is sought in a new frame within a radius of the previous position, proceeding sequentially. This is done by evaluating the pupil net convolutionally, as well as searching at five possible scales in order to track the object through scale space. The approach is described in more detail in Bertinetto et al. [1].

**Results and discussion.** Tab. 2 compares the methods in terms of the official metrics (accuracy and number of failures) for the VOT 2015 benchmark [11]. The ranking plot produced by the VOT toolkit is provided in the supplementary material (fig. B.1). From tab. 2, it can be observed that factorizing the filters in the siamese architecture significantly diminishes its performance, but using a learnet to predict the filters in the factorization recovers this gap and in fact achieves better performance than the original siamese net. The performance of the learnet architectures is not adversely affected by using the slimmer prediction networks B and C (with less channels).

An elementary tracker based on learnet compares favourably against recent tracking systems, which make use of different features and online model update strategies: DAT [17], DSST [3], MEEM [24], MUSTer [7] and SO-DLT [23]. SO-DLT in particular is a good example of direct adaptation of standard batch deep learning methodology to online learning, as it uses SGD during tracking to fine-tune an ensemble of deep convolutional networks. However, the online adaptation of the model comes at a big computational cost and affects the speed of the method, which runs at 5 frames-per-second (FPS) on a GPU. Due to the feed-forward nature of our one-shot learnets, they can track objects in real-time at framerates in excess of 60 FPS, while achieving less tracking failures. We consider, however, that our implementation serves mostly as a proof-of-concept, using tracking as an interesting demonstration of one-shot-learning, and is orthogonal to many technical improvements found in the tracking literature [11].

## 4 Conclusions

In this work, we have shown that it is possible to obtain the parameters of a deep neural network using a single, feed-forward prediction from a second network. This approach is desirable when iterative methods are too slow, and when large sets of annotated training samples are not available. We have demonstrated the feasibility of feed-forward parameter prediction in two demanding one-shot learning tasks in OCR and visual tracking. Our results hint at a promising avenue of research in "learning to learn" by solving millions of small discriminative problems in an offline phase. Possible extensions include domain adaptation and sharing a single learnet between different pupil networks.

**Acknowledgements**

This research was supported by Apical Ltd. and ERC grants ERC-2012-AdG 321162-HELIOS, HELIOS-DFR00200 and "Integrated and Detailed Image Understanding" (EP/L024683/1).

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
