[Supplementary Material]

# Learning feed-forward one-shot learners
# Supplementary material

**Luca Bertinetto**[*]
University of Oxford
luca@robots.ox.ac.uk

**João F. Henriques**[*]
University of Oxford
joao@robots.ox.ac.uk

**Jack Valmadre**[*]
University of Oxford
jvlmdr@robots.ox.ac.uk

**Philip H. S. Torr**
University of Oxford
philip.torr@eng.ox.ac.uk

**Andrea Vedaldi**
University of Oxford
vedaldi@robots.ox.ac.uk

## A   Basis filters

This appendix provides an additional interpretation for the role of the predicted filters in a factorized convolutional layer (Section 2.3).

To make the presentation succint, we will use a notation that is slightly different from the main text. Let $x$ be a tensor of activations, then $x_i$ denotes channel $i$ of $x$. If $a$ is a multi-channel filter, then $a_{ij}$ denotes the filter for output channel $i$ and input channel $j$. That is, if $a$ is $m \times n \times p \times q$ then $a_{ij}$ is $m \times n$ for $i \in [p]$, $j \in [q]$. The set $\{1, \ldots, n\}$ is denoted $[n]$.

The factorised convolution is

$$y = Ax = M'WMx \ . \tag{1}$$

where $M$ and $M'$ are pixel-wise projections and $W$ is a diagonal convolution. While a general convolution computes

$$(Av)_i = \sum_j a_{ij} * v_j \tag{2}$$

where each $a_{ij}$ is a single-channel filter, a diagonal convolution computes

$$(Wv)_i = w_i * v_i \tag{3}$$

where each $w_i$ is a single-channel filter, and a pixel-wise projection computes

$$(Mv)_i = \sum_j m_{ij} v_j \tag{4}$$

where each $m_{ij}$ is a scalar.

Let $q$ be the number of channels of $x$, let $p$ be the number of channels of $y$ and let $r$ be the number of channels of the intermediate activations. Combining the above gives

$$(WMx)_k = w_k * \left( \sum_{j \in [q]} m_{kj} x_j \right) = \sum_{j \in [q]} m_{kj} w_k * x_j \tag{5}$$

$$(M'WMx)_i = \sum_{k \in [r]} m'_{ik} \left( \sum_{j \in [q]} m_{kj} w_k * x_j \right) = \sum_{j \in [q]} \left( \sum_{k \in [r]} m'_{ik} m_{kj} w_k \right) * x_j \ . \tag{6}$$

This is therefore equivalent to a general convolution $y = Ax$ where each filter $a_{ij}$ is a combination of $r$ single-channel basis filters $w_k$

$$a_{ij} = \sum_{k \in [r]} m'_{ik} m_{kj} w_k \tag{7}$$

The predictions used in the dynamic convolution (Section 2.3) essentially modify these $r$ basis filters.

---

[*]The first three authors contributed equally, and are listed in alphabetical order.

# B  Additional results on object tracking

Table 1: Architectures are grouped by size of the main network. In addition to the official measures of the VOT toolkit, we report validation and training error for two measures that we use, together with the objective, to monitor the training phase. The displacement error measures the average euclidean distance between the peak of the output of the cross-correlation layer (the response) and the ground truth. The classification error, instead, expresses the likelihood that a random positive pair presents a response magnitude that is higher than the one of a random negative pair. For each group, the best entry for each column is in bold. Best overall entries are underlined.

| Architecture | Validation (training) error | | | VOT2015 scores | |
|---|---|---|---|---|---|
| Variant | Displacement | Classification | Objective | Accuracy | Failures |
| Siamese ($\varphi$=B) | 7.40 (6.26) | 0.426 (0.0766) | 0.156 (0.0903) | 0.465 | 105 |
| Siamese ($\varphi$=B; unshared) | 9.29 (6.95) | 0.514 (0.120) | **0.137** (0.0910) | 0.447 | 131 |
| Siamese ($\varphi$=B; factorised) | 8.58 (7.85) | 0.564 (0.160) | 0.141 (0.104) | 0.444 | 138 |
| Siamese learnet ($\varphi$=B; $\omega$=A) | 7.19 (**$\underline{5.86}$**) | 0.356 (0.0627) | **0.137** (0.0763) | **$\underline{0.500}$** | **$\underline{87}$** |
| Siamese learnet ($\varphi$=B; $\omega$=B) | **$\underline{7.11}$** (5.89) | **0.351** ($\underline{\mathbf{0.0515}}$) | 0.141 (**$\underline{0.0762}$**) | 0.497 | 93 |
| Siamese ($\varphi$=C) | 8.13 (7.5) | 0.589 (0.192) | 0.157 (0.112) | 0.466 | 120 |
| Siamese ($\varphi$=C; factorised) | 9.80 (8.96) | 0.539 (0.277) | 0.141 (0.117) | 0.435 | 132 |
| Siamese learnet ($\varphi$=C; $\omega$=A) | 7.51 (**6.49**) | 0.389 (**0.0863**) | $\underline{\mathbf{0.134}}$ (0.0856) | 0.483 | **105** |
| Siamese learnet ($\varphi$=C; $\omega$=C) | **7.47** (6.96) | $\underline{\mathbf{0.326}}$ (0.118) | 0.142 (0.0940) | **0.491** | 106 |

Figure 1: Accuracy-Robustness ranking plot (as produced by the VOT toolkit) for all the 62 trackers that participated to the VOT 2015 [1] challenge. Note that these rankings are produced based on a statistical method described in [1], and being relative rankings they are not comparable across papers; for this reason we supply the raw (absolute) metrics in the main paper. We use the variant B+A of our proposed (siamese) learnet. Best trackers are closer to the top right corner of the plot. Despite being only a proof-of-concept without online update of the model, our method is among the best. We remark that the top-ranking tracker, MDNet [2], fine-tunes a network with SGD during tracking, which is computationally expensive (1 frame per second on a GPU), while all our networks operate in feed-forward mode during tracking and run at at least 60 FPS. Moreover, MDNet is trained on videos from other benchmarks that are very similar to the ones of the test set, practice which has since been prohibited by the VOT committee.

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

| Initialization | Frame 100 | Frame 200 | Frame 400 |

Figure 2: Bounding box outputs of our tracker using the variant siamese learnet B+A. Even if our method uses exclusively the first frame as exemplar and does not perform any form of online update, it is robust to challenging tracking situations like change of appearance, motion blur and scale change. The snapshots have always been generated from frames 1, 100, 200 and 400. All the sequences belong to the VOT15 benchmark. From top to bottom: *iceskater2*, *basketball*, *car1*, *girl*, *helicopter*, *gymnastics1*, *road* and *pedestrian2*.