[Reviews · NeurIPS 2016]

Reviewer 1

Summary

The paper studies the problem of quickly adapting a network for the recognition of a new class based on a single example. The authors propose to parameterize net layers, both fully connected ones and convolutional ones, and train a second network, called learnet, to produce the parameters of the original one. The proposed parameterization is a factorization of each layer as a product of three matrices where the middle one is diagonal. The parameters are the values of the diagonal matrix. These parameters are predicted. The approach is studied on two tasks: one shot learning as defined by the Omniglot dataset; adaptive tracking of objects in videos.

Qualitative Assessment

The paper would benefit from baseline comparison on Omniglot. Also, a more thorough literature survey is needed. The idea of predicting net parameters, not surprisingly, is not born in the last few years: Schmidhuber: Learning to control fast-weight memories: An alternative to recurrent nets, 1992.

Confidence in this Review

3-Expert (read the paper in detail, know the area, quite certain of my opinion)


Reviewer 2

Summary

This paper introduces an end-to-end training framework for one-shot modulation of the parameters in a deep network. A small fraction of parameters in a convolutional layer is predicted from one class exemplar with the objective of turning the entire deep model into a same-different classifier. Performance is evaluated on competitive tasks such as Omniglot classification and single-target object tracking (VOT challenge 2015). The proposed framework is compared to several baseline on two competitive tasks and datasets. The results of extensive hyper-parameter searches for baselines and new method are presented without some confidence estimate, but this may be standard practice for those particular tasks.

Qualitative Assessment

Clarity: Although relatively straightforward, the proposed method is explained at length, and the paper includes a detailed account of the experimental methods; this make the paper easy to read. Originality: The paper introduces a particular incarnation of classic multiplicative-interactions ideas (e.g. 1, 2), ingeniously using a ‘learning-to-learn’ approach to quickly infer some parameters of a fast (feedforward) predictor at test time. Factored linear and convolutional layers have been used in a technically innovative way to reduce the size of the predicted parameter space, but are not new (e.g. 3, 4). Significance: The proposed method does not improve state-of-the-art results on the VOT 2015 challenge. The authors claim the new method to be more than an order of magnitude faster at test time than state-of-the-art methods, while providing comparable performance. This would put it in a class of its own, but the claim is not backed up with a comprehensive cost analysis.

Confidence in this Review

3-Expert (read the paper in detail, know the area, quite certain of my opinion)


Reviewer 3

Summary

This paper focuses on the problem of one-shot learning, where one wants to build a new classifier given just a single training image. The obvious complication in this problem is that with just one image it is very hard to predict whether the resulting network can generalize well in new categories, as the danger of overfitting is very high. To deal with this problem the authors propose to learn a second "meta"-network, whose outputs will be the parameters of the classification network. As the potential number of parameters to be returned is very high, especially given the number of available training images, the authors propose to factorize the matrix-vector multiplications of a fully connected in a form (M * diag(W) * M') and learn only a diagonal matrix W, while having M and M' pre-trained. Similarly, for the convolutional layers a filter basis is learnt and at test time it is decided which filter should be applied on which channel. Three different architectures are presented: a) a siamese architecture casting the one-shot classification as an image comparison problem, b) a siamese learnet, which predicts the shared parameters for some layers in a siamese network that compares images, and c) a learnet, which directly returns the parameters for the (not siamese) classification network. The method is evaluated on character recognition and on tracking.

Qualitative Assessment

The paper overall looks interesting with some nice ideas. Below I make a list of positive and negative elements and some remarks. Positives --------------- + The proposed method and the factorization makes sense and is interesting. Could one explain the matrices M and M' as learning some latent, topic models and then using w(z) to select the right proportions of these topic models for the novel category? Alternatively, could there be a connection with sparce coding, where at test time one needs to recover the right set of basis vectors? + The results support the method. In two benchmarks the obtained accuracies improve over reasonable (but not neceessarily the latest state-of-the-art) baselines. + The writing is mostly clear, except for section 2.3 (see below). Negatives & Remarks ---------------------- - The paper seems to overclaim and fails to cite some relevant papers. First and most importantly, there are some previous works, which also learn parameters dynamically, such as "A Dynamic Convolutional Layer for Short Range Weather Prediction" presented in CVPR 2015 and "DPPnet: Image Question Answering using Convolutional Neural Network with Dynamic Parameter Prediction" in arXiv, 2015. Although the first paper is not for one-shot learning, for image question answering the model also needs to learn in one-shot. Also, regarding object tracking the proposed method seems very close to "Siamese Instance Search for Tracking", CVPR 2016. The paper needs to make more precise statements and incorporate missing references. - Given that one wants discriminative classifiers, wouldn't it make more sense to consider triplet siamese networks that consider positive and negative pairs at the same time? How would such a network work compare to the standalone learnet? - Some clearer explanation would be welcome is in Section 2.3, where the factorized convolutional layers are explained. What is it exactly meant by saying that "M and M' disentangle the feature channels, allowing w(z) to choose which filter to apply to each channel"? Is there an over-redundant number of channels, which is then reshuffled and the right set of channels are selected to be convolved with the right filters? Also, it is not clear why the factorization of the convolutional layers should be convolutional by itself? Namely, the convolution operates as normal between filters F and the input x. Why does F need to also rely on some convolutional operation as well? Could one directly predict the parameters of the convolutional filters? - How is the weighted l1 distance in Table 1 computed?

Confidence in this Review

3-Expert (read the paper in detail, know the area, quite certain of my opinion)


Reviewer 4

Summary

This paper proposes a method that learns a neural network to predict the parameters of another network. Some technical details, e.g, how to reduce the number of parameters to be learnt, is also investigated. Experiments on two datasets verify the feasiblity of this idea.

Qualitative Assessment

In my opinion, this paper relates more to the topic of "learning to learn" than to "one-shot learning". Actually sometimes you have only one single training sample, and in that case, the proposed method will fail due to the lack of the outer "validation" sample. A reference is missed: "Learning to learn by gradient descent by gradient descent". The paper learns a neural network to teach another neural network to learn their parameters, instead of learns a parameter function for it as done in this paper. But obviously it is closely related to the topic discussed here. The experiments only test the situation of learning parameter predictors for only layer, It will be interesting to add some discussions on how to extend this to muliple layers. I also would like to see some comparisons with the methods from the line of "learning to learn".

Confidence in this Review

2-Confident (read it all; understood it all reasonably well)


Reviewer 5

Summary

This paper proposes a method to learn the parameters of a deep network in one shot by constructing a second deep network (learnet) which is capable to predict the parameter of a pupil network from a single exemplar. The proposed method is tested on the Omniglot dataset and Visual Object Tracking benchmark against siamese networks.

Qualitative Assessment

In my opinion, the paper is well written, clear, and the problem considered is of great importance. In the context in which deep networks need in general a lot of labeled training data, the capability of being able to learn in one-shot from a single example the deep network parameters would help improving various application domains, such as object tracking (as showed by the authors). In the experimental section I am a bit concerned on how the method would perform (or how it has to be adapted) if, by example, in the Omniglot dataset experiment the characters under scrutiny appear multiple times in the testing set of 20 characters and not just once, as it is now. Also, please could you comment on how the method tackle the same object at various sizes (e.g. when the object is increasing its size because it comes closer to the camera)? I saw that you consider 5 scales for object sizes, but I am not sure that these are sufficient? If accepted, I would recommend to the authors to proof read the paper once more. For instance, on line 248 maybe you could replace "alternative architectures" with "alternative siamese architectures". While doing this, I believe that it would be nice to capitalize the corresponding letters when the acronyms are introduced.

Confidence in this Review

1-Less confident (might not have understood significant parts)


Reviewer 6

Summary

Authors propose a very interesting way to approach single shot learning using deepnets. The propose learnet which takes in an example of the class and outputs parameters for a deepnet. The convolution and linear layers have factorized representations to make it feasible for the learnet to spit out the weights. The results on omniglot and visual object tracking look good both quantitatively and qualitatively. Overall, it's a novel way of looking at the problem and is a pretty good start in the direction of deepnets learning parameters for other networks which I have not seen before.

Qualitative Assessment

Would have been great if the others applied the one-shot learning for more classification tasks like imagenet etc. Also, supplementary material with tracking output would have been more convincing.

Confidence in this Review

2-Confident (read it all; understood it all reasonably well)